# Heterogeneity in establishment of polyethylene glycol-mediated plasmid transformations for five forest pathogenic *Phytophthora* species

**Erika N. Dort**[1]*, **Richard C. Hamelin**[1,2,3]

**1** Department of Forest & Conservation Sciences, Faculty of Forestry, University of British Columbia, Vancouver, British Columbia, Canada, **2** Institut de Biologie Intégrative et des Systèmes (IBIS), Université Laval, Québec City, Québec, Canada, **3** Département des Sciences du bois et de la Forêt, Faculté de Foresterie et Géographie, Université Laval, Québec City, Québec, Canada

* edort@alum.ubc.ca

## Abstract

Plasmid-mediated DNA transformation is a foundational molecular technique and the basis for most CRISPR-Cas9 gene editing systems. While plasmid transformations are well established for many agricultural *Phytophthora* pathogens, development of this technique in forest Phytophthoras is lacking. Given our long-term research objective to develop CRISPR-Cas9 gene editing in a forest pathogenic *Phytophthora* species, we sought to establish the functionality of polyethylene glycol (PEG)-mediated plasmid transformation in five species: *P. cactorum*, *P. cinnamomi*, *P. cryptogea*, *P. ramorum*, and *P. syringae*. We used the agricultural pathogen *P. sojae*, a species for which PEG-mediated transformations are well-established, as a transformation control. Using a protocol previously optimized for *P. sojae*, we tested transformations in the five forest Phytophthoras with three different plasmids: two developed for CRISPR-Cas9 gene editing and one developed for fluorescent protein tagging. Out of the five species tested, successful transformation, as indicated by stable growth of transformants on a high concentration of antibiotic selective growth medium and diagnostic PCR, was achieved only with *P. cactorum* and *P. ramorum*. However, while transformations in *P. cactorum* were consistent and stable, transformations in *P. ramorum* were highly variable and yielded transformants with very weak mycelial growth and abnormal morphology. Our results indicate that *P. cactorum* is the best candidate to move forward with CRISPR-Cas9 protocol development and provide insight for future optimization of plasmid transformations in forest Phytophthoras.

## Introduction

The oomycete genus *Phytophthora*, comprising almost 200 species, is well-known for containing an abundance of destructive plant pathogens [1]. The substantial economic and environmental impacts of *Phytophthora* species in agriculture, forestry, and horticulture have led to

**Funding:** This work was supported by a Genome Canada Large-Scale Applied Research Project (https://genomecanada.ca/) awarded to RH (and co-applicants) under Grant #10106 (bioSAFE: Biosurveillance of Alien Forest Enemies). We also acknowledge the support of a Natural Sciences and Engineering Research Council of Canada (https://www.nserc-crsng.gc.ca/index_eng.asp) CGS-D fellowship awarded to ED. The funders had no role in study design, data collection and analysis, decision to publish, or preparation of the manuscript.

**Competing interests:** The authors have declared that no competing interests exist.

decades of research focused on improved management and mitigation of the damage caused by these ubiquitous plant pathogens. The continued development of molecular DNA techniques and sequencing technologies in the last 20–30 years has shifted *Phytophthora* research increasingly towards genetics and genomics studies with the intent of elucidating the genetic mechanisms that drive phytopathogenicity and other related traits [2–7]. The development of CRISPR-Cas9 gene editing [8] and related technologies has provided new tools for the genetic exploration of phytopathogenicity, and CRISPR-Cas9 protocols have now been developed for several agricultural *Phytophthora* pathogens [9–13]. CRISPR-Cas gene editing technologies have proven to be invaluable in elucidating the genetic mechanisms driving *Phytophthora*-host interactions and phytopathogenicity [9, 10, 14, 15], and exploring fungicide resistance [11, 13, 16]. However, CRISPR-Cas9 gene editing has yet to be established in any forest pathogenic Phytophthoras, likely due to the lack of stable and efficient DNA transformation protocols in these species. Generating long-term management strategies for forest *Phytophthora* pathogens will require a deeper understanding of the genetics driving their phytopathogenicity; however, that will not be possible without the development of reliable molecular techniques. Establishing DNA transformations in forest Phytophthoras is therefore a crucial step to bring these species into the molecular genetics era and will allow the development of exploratory techniques such as CRISPR-Cas9 gene editing.

DNA transformation is a fundamental molecular technique that is used to study cellular processes and characterize specific genes involved in pathways or interactions of interest. The development of CRISPR-Cas9 protocols and similar techniques for any organism relies on a functioning and reliable DNA transformation system. The first *Phytophthora* species in which oomycete-specific transformation plasmids were developed was *P. infestans* (Mont.) de Bary, the causal agent of late blight of potato [17, 18]. The plasmids constructed by Judelson and colleagues for these early transformation studies were derived from the common plasmid cloning vector pUC19 [19], modified to include promoter and transcription terminator sequences from the *ham34* and *hsp70* genes of the oomycete *Bremia lactucae* Regel [17, 18]. Neomycin phosphotransferase (*nptII*) and hygromycin phosphotransferase (*hpt*) genes were used to generate stable *P. infestans* transformants resistant to geneticin (G418) or hygromycin B, respectively [18]. These first *Phytophthora* transformations used PEG and calcium chloride ($CaCl_2$) to transform cell-wall digested protoplasts [17, 18], employing a protocol adapted from transformations successfully developed in fungi [20].

In the following decades, the PEG-$CaCl_2$-mediated transformation approach was adapted and modified for other *Phytophthora* species, including *P. sojae* Kaufm. & Gerd [21], *P. nicotianae* Breda de Haan [22], *P. palmivora* (E.J. Butler) E.J. Butler [23], *P. brassicae* De Cock & Man in't Veld [24], *P. citricola* Sawada [25], *P. capsici* [26], and *P. cactorum* (Lebert & Cohn) J. Schröt [27]. Alternative transformation methods were also developed for *Phytophthora* species including microprojectile bombardment [28, 29], electroporation [30, 31], and *Agrobacterium*-mediated transformation [32, 33]. Each transformation method has its own strengths and weaknesses. However, the PEG-$CaCl_2$ method remains the most popular due to its proven success with various *Phytophthora* species and its flexibility, making it suitable for research labs that lack highly specialized equipment [25, 34].

While molecular protocols have been well-developed for agricultural Phytophthoras, research is lacking for forest Phytophthoras. Only four studies have reported DNA transformation systems in forest pathogenic *Phytophthora* species: two using the original PEG-$CaCl_2$ protocol developed by Judelson *et al.* [17] to transform the jarrah dieback pathogen *P. cinnamomi* Rands [35] and the sudden oak death pathogen *P. ramorum* Werres, De Cock, & Man in't Veld [36]; and two performing the optimized PEG-$CaCl_2$ transformation [25] in *P. cinnamomi* [37] and the soilborne pathogen *P. cactorum* [27]. However, there have been no follow-up

studies in either *P. ramorum* or *P. cactorum* after their initial transformation, so the long-term reliability and reproducibility of DNA transformations in these species and other forest Phytophthoras remains unknown. The objective of this study was to test the optimized PEG-CaCl$_2$ transformation [25, 38, 39] protocol in five forest *Phytophthora* pathogens: *P. cactorum*, *P. cinnamomi*, *P. cryptogea* Pethybr. & Laff., *P. ramorum*, and *P. syringae* (Kleb.) Kleb. The purpose of testing five species was two-fold: first, to establish the adaptability and reproducibility of PEG-CaCl$_2$ transformations across common forest pathogenic *Phytophthora* species; and second, to determine which *Phytophthora* forest pathogens can be readily transformed and would therefore be good candidates for CRISPR-Cas9 gene editing protocols.

## Materials and methods

Note: the detailed protocols used for all growth media and transformation solutions are provided in S1 Appendix.

### *Phytophthora* species, isolates, and culture conditions

The *Phytophthora* species and isolates used in this study are summarized in Table 1. All cultures were maintained on clarified 20% V8 agar (V8A: see S1 Appendix) at room temperature (~21˚C) in the dark. For long-term storage, agar plugs taken from the edge of mycelial growth, five days post plating, were stored in nuclease-free molecular water at 5–10˚C in the dark.

### Transformation plasmids

Three plasmids were tested for transformations: pYF2-PsNLS-hSpCas9-GFP (abbreviated pYF2-PsCG), pYF515, and pGFPN. Both pYF2-PsCG and pYF515 are CRISPR-Cas9 plasmids developed for *P. sojae*. pYF2-PsCG was originally developed to test expression of a human codon-optimized *Streptococcus pyogenes* Cas9 (hSpCas9) gene in *P. sojae*; the hSpCas9 gene is fused at its 5' end to a nuclear localization sequence from *P. sojae* (PsNLS) and tagged at its 3' end with a green fluorescent protein (GFP) [9, 42]. The pYF2-PsCG plasmid also contains genes for ampicillin resistance (AmpR: conferred by the β-lactamase gene from the pUC19 plasmid) and geneticin resistance (*nptII*) (S1 Fig). pYF515 was developed as an 'all-in-one' transformation plasmid to express both hSpCas9 and a single-guide RNA (sgRNA) for CRISPR-Cas9 gene editing [38]. Like pYF2-PsCG, pYF515 has AmpR and *nptII* antibiotic resistance genes (S2 Fig). pGFPN is a transformation plasmid developed for fluorescent protein tagging in *Phytophthora* species [43] and contains a GFP gene in addition to the AmpR and *nptII* antibiotic resistance genes (S3 Fig). The DNA sequences of all three transformation plasmids are provided in S1–S3 Files.

### Plasmid sequencing

DNA from all three plasmids was transformed into DH5α *Escherichia coli* chemically competent cells (Thermo Fisher Scientific, Waltham, MA, USA) for replication. Plasmid DNA was extracted using either a QIAGEN HiSpeed® Plasmid Midi Kit or Maxi Kit (QIAGEN Sciences, Germantown, MD, USA) as per the manufacturer's instructions (HiSpeed® Plasmid Purification Handbook). The extracted DNA was adjusted to a concentration of 30 ng/µL for sequencing. Whole plasmid sequencing was performed by Plasmidsaurus (Eugene, OR, USA) using Oxford Nanopore Technology with custom analysis and annotation. Note that two independent preparations of the pGFPN plasmid were sent for sequencing due to issues experienced with this plasmid in the lab. The sequence reads returned from Plasmidsaurus for each plasmid preparation were mapped to the respective expected sequence using Geneious Prime

**Table 1. Isolate details for the six *Phytophthora* species used in this study.**

| Species | Isolate | Isolate Details |
|---|---|---|
| *Phytophthora cactorum* | Larch FF-42 2Pa | See Feau *et al.* 2021 [40] for isolate details. |
| *Phytophthora cinnamomi* | CBS 270.55 | Purchased from the CBS-KNAW culture collection (Westerdijk Fungal Biodiversity Institute). Originally isolated in 1954 by I.W. Buddenhagen from *Chamaecyparis lawsoniana* var. *alumii* in Boskoop, Netherlands. |
| | JP-09-065 (strain A-3/3/09-P-2-st) | Isolated in 2009 by J. Parke from *Pieris japonica* in Marion, OR, USA. Culture provided by N. Grünwald. |
| *Phytophthora cryptogea* | RS-2006-soil-d8 | Sample collected in 2017 by N. Feau from the soil surrounding *Pinus monticola* on Vancouver Island, BC, Canada. Culture isolated by W. Vasquez and provided by N. Feau. |
| *Phytophthora ramorum* | NA2 17_0134_0030 | Cultures of both isolates provided by G. Bilodeau. |
| | NA2 16_0386_0016 | |
| *Phytophthora sojae* | P6497 (race 2) | Isolated from infected *Glycine max* plants in Mississippi, USA. Culture provided by F. Arredondo. |
| *Phytophthora syringae* | Alder 12 | See Feau *et al.* 2022 [41] for isolate details. |

The five forest pathogenic *Phytophthora* species are *P. cactorum*, *P. cinnamomi*, *P. cryptogea*, *P. ramorum*, and *P. syringae*. The agricultural pathogen *P. sojae* was included as a control species for plasmid transformations.

2023.2.1 (https://www.geneious.com) to determine the percent identity of each plasmid relative to its reference sequence.

## Determining the minimum inhibitory concentrations of geneticin for *Phytophthora* isolates

The antibiotic geneticin (G418) was used for selection of *Phytophthora* transformants. Prior to transformations, growth assays for all six *Phytophthora* species used in this study were performed on V8A plates supplemented with a gradient of G418 (Gibco™ Geneticin™ Selective Antibiotic, Thermo Fisher Scientific, Waltham, MA, USA) concentrations to determine the minimum inhibitory concentration (MIC) for each species. The G418 concentrations tested for each species and isolate are listed in S1 Appendix. The volume of medium per plate was standardized and two replicate plates were produced for each G418 concentration. The MIC for each species was recorded as the first G418 concentration at which mycelial growth stopped (see example for *P. cactorum* in S4 Fig).

## Protoplast isolation and transformation

An optimized PEG-mediated protoplast transformation protocol [38], based on previously described methods [25, 39], was used to introduce plasmid DNA into *Phytophthora* cells. Beyond testing the three different plasmids, various modifications were made for culture media, enzyme digestion times, plasmid DNA concentrations, protoplast recovery media, and G418 selection concentration to test their effects on transformation success in the five different forest *Phytophthora* species (see S1 Table for all conditions tested).

## Preparation of *Phytophthora* cultures for transformation

*Phytophthora* cultures were initially grown on 1.5% agar plates (nutrient pea agar [NPA], nutrient V8 agar [NVA], or 20% V8 agar [V8A]) at room temperature (~21°C) in the dark. Once there was sufficient mycelial growth (five to seven days), cultures were transferred to a liquid medium (nutrient pea broth [NPB], nutrient V8 broth [NVB], or V8 broth [V8B]):

three flasks, each containing 50 mL of broth, were inoculated with five agar plugs (6-mm diameter) taken from the growing edge of the mycelial culture (15 agar plugs in total). The liquid cultures were grown for three days at room temperature (~21˚C) in the dark.

**Isolation of protoplasts.** After three days of growth, mycelial mats were harvested and rinsed with autoclaved Milli-Q® water followed by 0.8 M mannitol. The mycelial mats were transferred to a 100 mm × 20 mm Petri dish, covered with 0.8 M mannitol, and shaken gently (~30 rpm) for 10 minutes. The mats were recollected, transferred to a new Petri dish containing 20 mL of an enzyme solution (lysing enzymes + cellulase; see S1 Appendix) and digested for 40–105 minutes (see S1 Table for isolate-specific conditions) at 25˚C with gentle shaking (~30 rpm). Protoplast release was monitored with microscopy. After digestion, the enzyme/protoplast solution was filtered through a 70 μm cell strainer and centrifuged at 1200 rcf for 2.5 minutes. The pellet was gently washed in 35 mL of W5 solution (see S1 Appendix), recentrifuged at 1200 rcf for 3 minutes, and the resulting pellet was resuspended in 10 mL of W5 solution and placed on ice for a minimum of 30 minutes. The protoplast concentration was determined using a hemocytometer.

**Transformation of protoplasts with plasmid DNA.** Following incubation on ice, the protoplast suspension was centrifuged at 1200 rcf for 4 minutes. The resulting pellet was resuspended in MMg solution (see S1 Appendix) at a concentration of $2 \times 10^6$– $2 \times 10^7$ protoplasts/mL and incubated at room temperature for 10 minutes. During incubation, $\leq$15 μL of plasmid DNA (13–55 μg, see S1 Table for amounts tested), or sterile molecular-grade water for the control treatment, was added to a 50 mL centrifuge tube and placed on ice. After the 10-minute incubation, the protoplasts were gently mixed, and 1 mL of the protoplast/MMg solution was added to the DNA (or water) in each 50 mL tube, mixed, and incubated for 30 minutes on ice. A 40% (v/v) PEG solution was added to each tube in three aliquots of 580 μL for a total of 1.74 mL of PEG solution per tube, and the tubes were incubated for another 30 minutes on ice. After 30 minutes, 2 mL of pea-mannitol (PM) or V8-mannitol (VM) broth was added to each tube. The tubes were inverted to mix and incubated on ice for 2 minutes. Then, 8 mL of PM/VM broth was added to each tube, the tubes were inverted to mix and incubated at room temperature for an additional 2 minutes. To test the viability of the protoplasts after the transformation process, ~30–50 μL of the protoplast-PM/VM solution from each tube (including control treatment) was spread onto an agar plate (NPA or NVA) and incubated at room temperature in the dark overnight (day one protoplast viability test). The remainder of the protoplast-PM/VM solution (~10 mL) from each tube was added to a 100 mm × 20 mm Petri dish containing 10 mL of PM/VM broth with 100 μg/mL of ampicillin (Thermo Fisher Scientific, Waltham, MA, USA) for a final concentration of 50 μg/mL ampicillin (20 mL total volume). The dishes were incubated overnight at room temperature in the dark.

**Antibiotic selection of transformants.** The following morning, the regenerated protoplasts were transferred to a new 50 mL tube and centrifuged for 5 minutes at 2000 rcf. The pellet was resuspended in ~2 mL of the supernatant and 150 μL of the resuspended protoplasts were added to 20 mL of cooled PM/VM agar (40–45˚C) without G418 and poured into a Petri dish (day two protoplast viability test). Cooled PM/VM agar (40–45˚C) with 5–50 μg/mL of G418 (see S1 Table for concentrations tested) was added to the remaining protoplasts to a total volume of 50 mL. The tubes were inverted to mix, and the protoplast/agar mixture was poured into three Petri dishes (~16–17 mL of mixture per dish). All plates were incubated at room temperature in the dark for two days, or until mycelial growth appeared (up to five days). Each mycelial colony that grew on the G418-supplemented medium was treated as a putative transformant and was transferred to a well of a 12-well culture plate containing 2 mL of V8 broth with an increased concentration of G418 (20–60 μg/mL, see S1 Table for concentrations tested).

**Table 2. Primer pairs used for diagnostic PCR to detect plasmid DNA in *Phytophthora* transformants.**

| Primer Pair Name | Primer Sequences (5' to 3') | | Annealing Temperature | Plasmids | Expected Product Size |
|---|---|---|---|---|---|
| eGFP_Diag_F1/R1 | F1: GGAACTGGATGGTGATGTGAAC | | 51.4°C | pGFPN, pYF2-PsCG | 274 bp |
| | R1: CTTGTAGTTCCCGTCATCCTTG | | | | |
| pYF_Cas9_Diag_F2/R2 | F2: GATGAACACTAAGTACGACGAG | | 60.0°C | pYF515, pYF2-PsCG | 731 bp |
| | R2: GAAAGTCGATGGGATTCTTCTC | | | | |
| pYF_Cas9_Diag_F3/R3 | F3: GCTGCCTGAGAAGTACAAAGAG | | 62.0°C | pYF515, pYF2-PsCG | 824 bp |
| | R3: CCAGAATGTCCTCGTTTTCCTC | | | | |

All primers were designed using NCBI Primer-BLAST [44] with target specificity checked against the "RefSeq representative genomes" database for Phytophthora (taxid: 4783) to minimize off-target amplification.

### Identification of successful transformants

Any putative transformants that grew through the second round of G418 selection were considered successful transformants based on their stable growth on selective medium. Mycelial tissue from stable transformants was harvested for DNA extraction and subsequent diagnostic PCR to confirm the presence of plasmid DNA. Wildtype (non-transformed) cultures were grown in parallel as a negative control. After 3–4 days of growth in V8 broth supplemented with G418 (no G418 for wildtype cultures), 50–100 mg of mycelial tissue was harvested from each culture, and flash frozen in liquid nitrogen. Total cellular DNA was extracted using a QIAGEN DNeasy® Plant Mini Kit (QIAGEN Sciences, Germantown, MD, USA) as per the manufacturer's instructions (DNeasy® Plant Handbook, Mini Protocol, TissueLyser procedure). Tissue disruption was performed using a Retsch Mixer Mill MM 300 (Retsch GmbH, Haan, Germany). PCR was performed on the DNA extractions from plasmid transformants, wildtype cultures (negative control), and pure plasmid DNA (positive control) using Phusion™ Plus DNA Polymerase (Thermo Fisher Scientific, Waltham, MA, USA) and primers (Integrated DNA Technologies, Coralville, IA, USA) designed to amplify specific regions of each plasmid (Table 2). All PCRs were performed using a Bio-Rad T100™ thermal cycler (Hercules, California, USA) and the following cycling conditions (as per manufacturer's instructions): an initial hold of 98°C for 30 seconds; 35 cycles of 98°C for 10 seconds, 60°C (universal annealing temperature used for Phusion™ Polymerase), for 10 seconds, 72°C for 30 seconds; a final hold of 72°C for 5 minutes. The resulting PCR products were analysed using gel electrophoresis. The presence of an amplified band at the appropriate size confirmed the transformation of plasmid DNA.

## Results

### Sequencing of *Phytophthora* transformation plasmids indicates instability of pGFPN

The sequencing results from Plasmidsaurus showed the expected sequences for pYF515 (99.8% pairwise identity to original plasmid sequence; S5A Fig) and pYF2-PsCG (99.8% pairwise identity to original plasmid sequence; S5B Fig), but not for either of the two pGFPN samples (52.9% and 62.6% pairwise identity to original plasmid sequence; S5C and S5D Fig, respectively). The expected size of the pGFPN plasmid was 6,891 bp, however, the sequencing results indicated that the two pGFPN samples were 9,842 bp and 4,927 bp with apparent gene repeats, deletions, and inversions relative to the original plasmid sequence (S5C and S5D Fig, respectively).

### Forest *Phytophthora* species differ in geneticin sensitivity

The five forest *Phytophthora* species tested in this study exhibited varied sensitivities to G418, with MICs ranging from 5 μg/mL to 40 μg/mL of G418 (Fig 1). *Phytophthora syringae* and *P. ramorum* were the most sensitive species with MICs of 5 μg/mL G418, *P. cactorum* and *P. cryptogea* were more resistant with MICs of 20 μg/mL G418, and *P. cinnamomi* was the most resistant with an MIC of 40 μg/mL G418. The MICs were used to customize the G418 concentration used for selection of plasmid transformants for each *Phytophthora* species (see S1 Table).

### Plasmid DNA transformations are inconsistent and unstable in forest *Phytophthora* species

Success of plasmid transformations, as determined by stable mycelial growth of transformants after a second round of selection on an increased concentration of G418, was inconsistent between the five forest *Phytophthora* species tested (Table 3, S1 Table). Control transformations in *P. sojae* with the pYF-PsCG plasmid always yielded stable transformants with positive diagnostic PCR results (Fig 2). Note that there are slight differences in band size in the diagnostic PCR (Fig 2), which was a problem we encountered consistently when amplifying products from the pYF2-PsCG plasmid even with different primer pairs. Sanger sequencing in both the forward and reverse directions of the PCR products from Fig 2 showed identical sequences

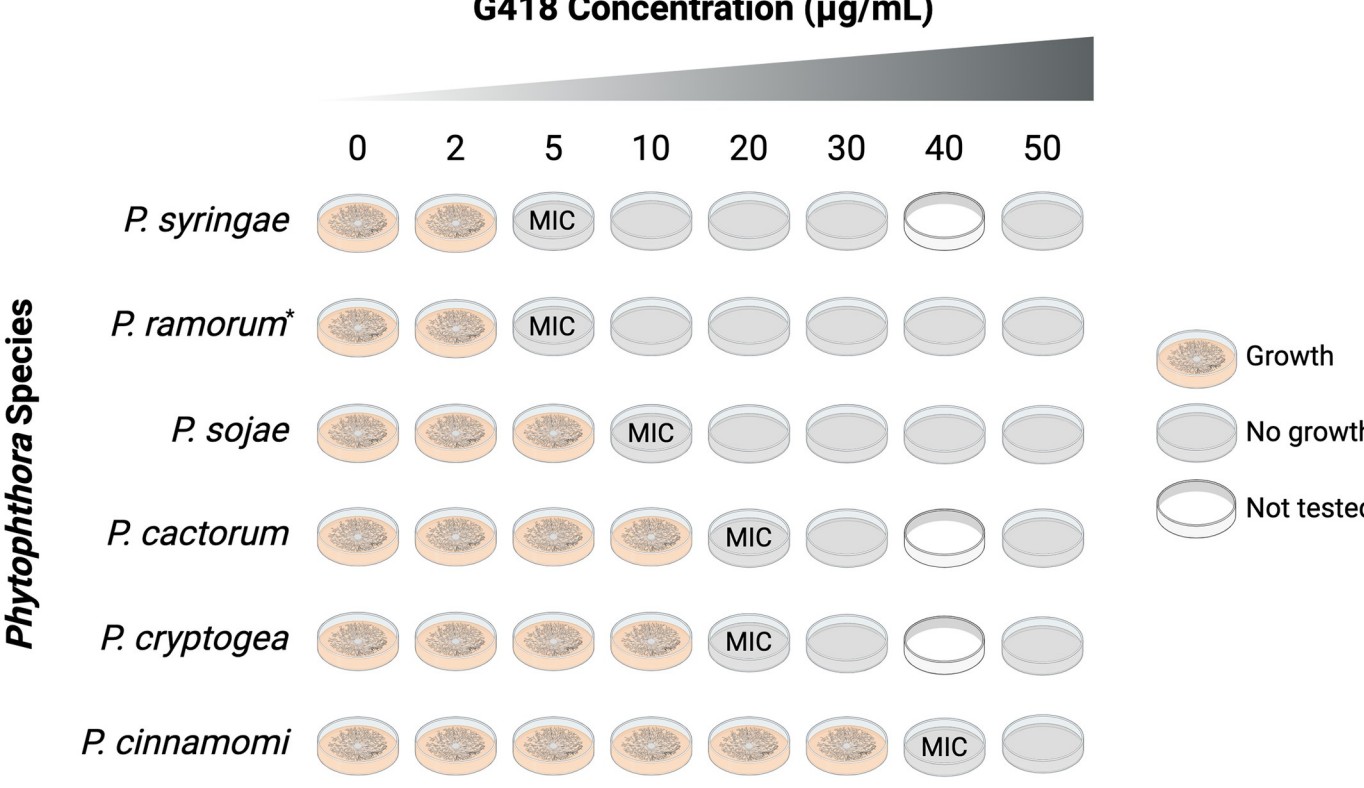

**Fig 1. Minimum inhibitory concentrations (MICs) of geneticin (G418) for *Phytophthora* cultures.** The G418 MIC was determined for each of the five forest *Phytophthora* species tested in this study and the agricultural pathogen *P. sojae* (used as a transformation control). Two replicates were tested for each species/isolate. *Two isolates of *P. ramorum* were tested, NA2 16_0386_0016 and NA2 17_0134_0030, and growth stopped for both at 5 μg/mL of G418. Figure created with BioRender.com (Agreement #MS26FYBSTT).

**Table 3. Summary of *Phytophthora* transformation results.**

| Species | Isolate | Growth Media Tested | Recovery Media Tested | Plasmid | Transformation Success[a] | Diagnostic PCR Confirmation[b] |
|---|---|---|---|---|---|---|
| *Phytophthora sojae* (control) | P6497 (R2) | NPA/NPB | PM | pYF2-PsCG | Yes | Yes |
| | | | | pGFPN | No | NA |
| *Phytophthora cactorum* | Larch FF-42 2Pa | NPA/NPB, NVA/NVB | PM, VM | pYF2-PsCG | Yes | Yes |
| | | | | pGFPN | No | NA |
| *Phytophthora cinnamomi* | CBS 270.55 | NPA/NPB | PM | pYF2-PsCG | No | NA |
| | JP-09-065 | NPA/NPB | PM | pGFPN | No | NA |
| *Phytophthora cryptogea* | RS-2006-soil-d8 | NPA/NPB | PM | pYF2-PsCG | No | NA |
| *Phytophthora ramorum* | NA2 17_0134_0030 | NPA/NPB, NVA/NVB, V8A/V8B | PM, VM | pYF2-PsCG | Yes | Yes |
| | | | | pGFPN | No | NA |
| | | | | pYF515 | Yes | Yes |
| *Phytophthora syringae* | Alder 12 | NPA/NPB, V8A/V8B | PM | pYF2-PsCG | No | NA |

The detailed summary of all conditions tested for transformations in each species as well as the results obtained are outlined in S1 Table. Abbreviations: NPA = nutrient pea agar, NPB = nutrient pea broth, NVA = nutrient V8 agar, NVB = nutrient V8 broth, V8A = V8 agar, V8B = V8 broth, PM = pea-mannitol medium, VM = V8-mannitol medium.

[a] As determined by stable growth of transformed cultures after increased G418 selection.

[b] As determined by generation of a plasmid amplicon from PCR

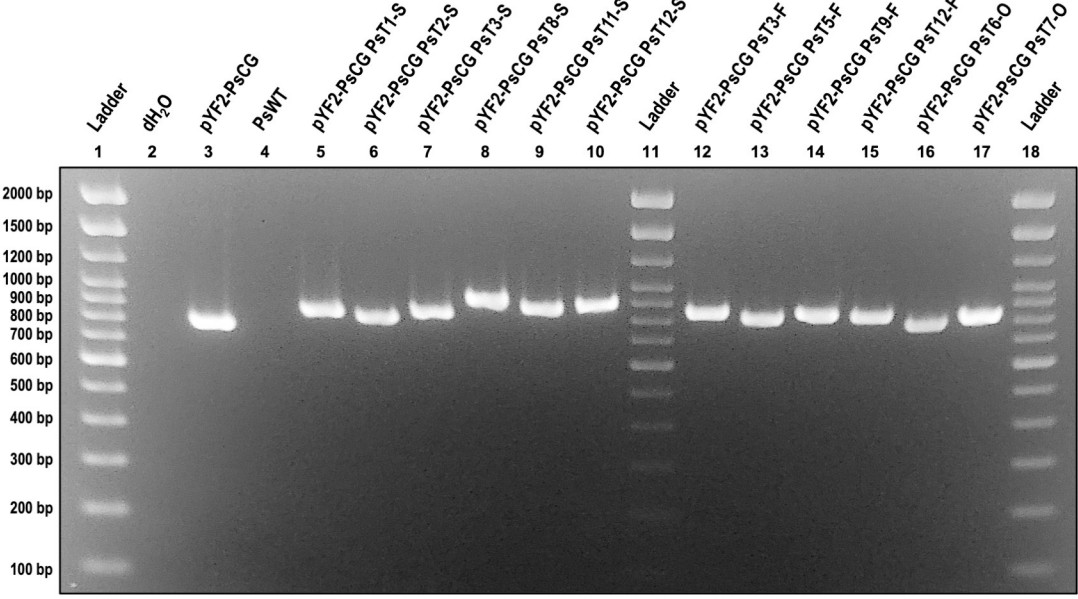

**Fig 2. Diagnostic PCR results for *Phytophthora sojae* wildtype and pYF2-PsCG transformants.** Gel electrophoresis of products from a PCR performed on DNA from *Phytophthora sojae* P6497 wildtype (PsWT; well 4) and 12 pYF2-PsCG transformant (pYF2-PsCG PsT; wells 5–10 and 12–17) cultures using the pYF_Cas9_Diag_F2/R2 primer pair (expected product size is 731 bp). The negative PCR control was a reaction using molecular-grade water instead of DNA (well 2). The positive PCR control was a pYF2-PsCG plasmid DNA extraction (wells 3). Note that the DNA used for the plasmid control was the same preparation that was used for both *P. sojae* transformations. The gel was run at 100 volts for 75 minutes on 1.75% agarose in 0.5X TBE buffer with a 100 bp DNA Ladder (Thermo Fisher Scientific, Waltham, MA, USA). All samples were loaded with Safe-Green™ stain (Applied Biological Materials Inc., Richmond, BC, Canada) at a ratio of 1 μL Safe-Green: 5 μL diluted PCR product.

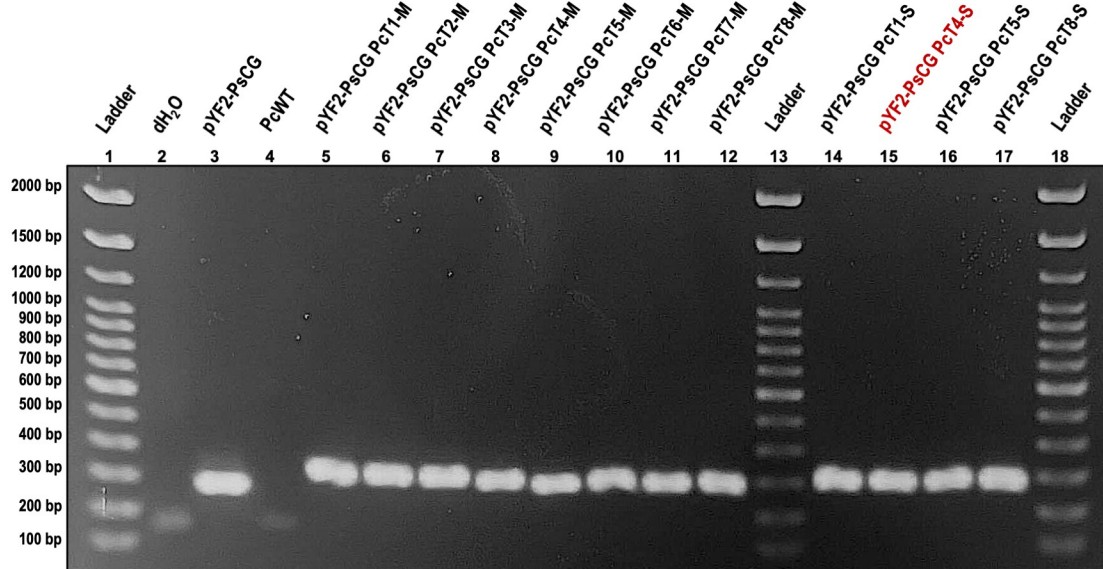

**Fig 3. Growth of *Phytophthora cactorum* wildtype and pYF2-PsCG transformant cultures.** Transformations of *Phytophthora cactorum* FF42 with the pYF2-PsCG plasmid yielded stable transformants. **(A)** No growth of wildtype (untransformed) cultures on V8 agar + 40 μg/mL G418 relative to **(B)** growth of transformant cultures on the same medium. Photos were taken five days post plating. **(C)** Transformants growing stably on V8 agar + 40 μg/mL G418 after multiple rounds of sub-culturing. Photos were taken ten days post plating.

to each other and to the positive control. Therefore, the band differences on the gel electrophoresis are likely due to secondary structures in the PCR products or are artifacts from the PCR reaction.

Other than the *P. sojae* control, the only species in which consistent and stable plasmid transformations could be reliably reproduced was *P. cactorum* isolate Larch FF-42 2Pa (hereafter abbreviated FF42). Transformations of *P. cactorum* with the pYF2-PsCG plasmid yielded stable transformants that, even after multiple rounds of sub-culturing, grew well on V8A plates supplemented with 40 μg/mL of G418 (Fig 3). The results from diagnostic PCR were also consistent for *P. cactorum*, with amplification even in transformants with reduced growth in the increased G418 concentration (Fig 4). Note that preliminary results with transformations

**Fig 4. Diagnostic PCR results for *Phytophthora cactorum* wildtype and pYF2-PsCG transformants.** Gel electrophoresis of products from a PCR performed on DNA from *Phytophthora cactorum* FF42 wildtype (PcWT; well 4) and pYF2-PsCG transformant (pYF2-PsCG PcT; wells 5–12 and 14–17) cultures using the eGFP_Diag_F1/R1 primer pair (expected product size is 274 bp). The negative PCR control was a reaction using molecular-grade water instead of DNA (well 2), and the positive PCR control was purified pYF2-PsCG plasmid DNA (well 3). The gel was run at 100 volts for 60 minutes on 1% agarose in 0.5X TBE buffer with a 100 bp DNA Ladder (Thermo Fisher Scientific, Waltham, MA, USA). All samples were loaded with Safe-Green™ stain (Applied Biological Materials Inc., Richmond, BC, Canada) at a ratio of 1 μL Safe-Green: 5 μL diluted PCR product. The pYF2-PsCG PcT4-S (well 15) transformant is highlighted in red because it struggled to grow through the second round of G418 selection (40 μg/mL), with roughly only 25% mycelial growth relative to the other three transformants in the same experiment (T1-S, T5-S, T8-S), however, a strong band amplified in the diagnostic PCR.

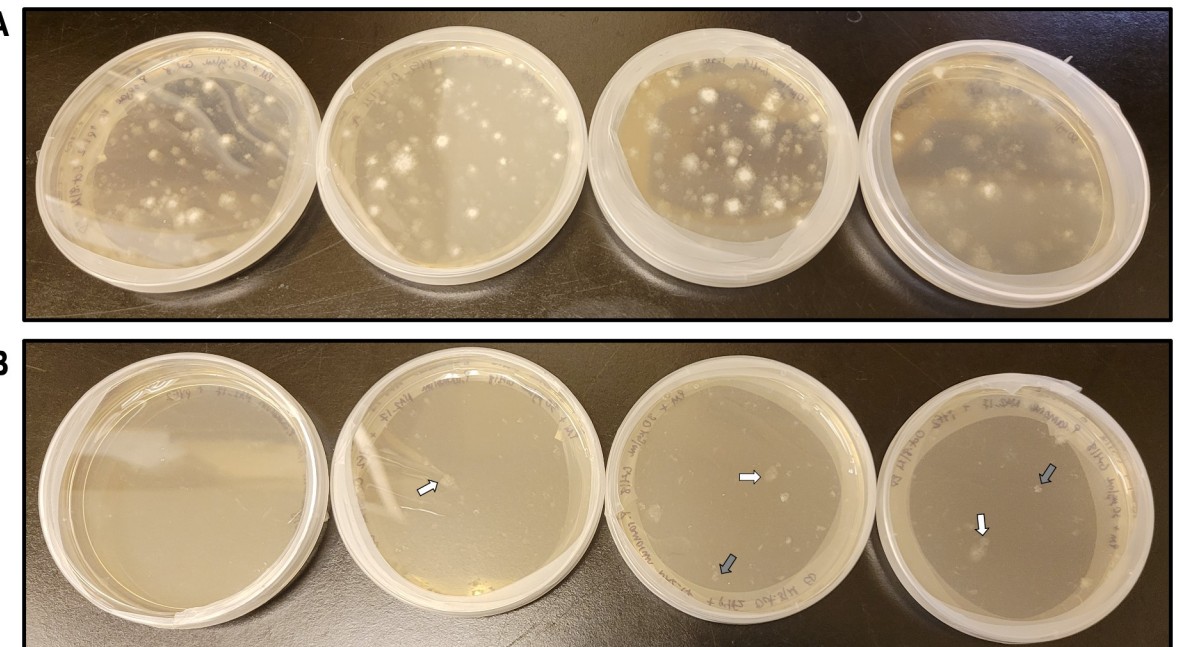

**Fig 5. Comparison of *Phytophthora sojae* and *P. ramorum* putative transformant mycelial growth.** Putative transformants (PTs) of the control species *Phytophthora sojae* P6497 (**A**) and *P. ramorum* NA2_17 (**B**) growing on V8 agar supplemented with 50 µg/mL G418 and 30 µg/mL G418, respectively, four days post transformation. While putative *P. ramorum* transformants did grow on three of the plates, the mycelial growth was weak and irregular (see arrows) relative to the *P. sojae* control. The white arrows indicate PTs that would have been considered "strong" for *P. ramorum*, whereas the grey arrows indicate very weak PTs. All *P. ramorum* PTs were taken through a second round of selection, but most did not grow at the increased concentration of G418; none of the "weak" PTs (grey arrows) grew, and only two of the "strong" PTs (white arrows) grew. The same pattern emerged for every *P. ramorum* transformation attempted.

using the pYF515 plasmid were also successful in *P. cactorum* (S1 Table), yielding transformants with stable growth after multiple rounds of G418 selection, however, diagnostic PCR was not performed, so the amplification of plasmid DNA from transformant cultures could not be confirmed.

We achieved some initial success with transformations of *P. ramorum* isolate NA2 17_0134_0030 (hereafter abbreviated NA2_17) with both pYF2-PsCG and pYF515. However, transformants were difficult to obtain and culture: the mycelial growth of putative transformants was weak and irregular relative to the *P. sojae* control transformants (Fig 5). Additionally, most of the putative transformants for *P. ramorum* that were transferred to the second round of increased G418 selection did not grow (S1 Table). However, the transformants that did grow in the second round of G418 selection all yielded positive results for the plasmid DNA diagnostic PCR (Fig 6). Note that similar to Fig 2, the amplified products from the pYF2-PsCG plasmid in Fig 6 exhibited differences in migration on the gel. Again, these differences were likely due to secondary structures or PCR artifacts, but it was still clear that there was plasmid amplification from the transformant cultures and not from the wildtype cultures. Notably, subsequent transformations with *P. ramorum* under the same conditions (i.e., same protocol, plasmid DNA, recovery medium, etc.) failed, with no putative transformants growing on G418 selective medium.

Once we established some transformation success in *P. cactorum* and *P. ramorum*, we also tested V8-based growth and recovery media in both species to determine whether it would have any effect on protoplast formation or transformation success (Table 3). Our results showed that the change in medium had no obvious effect for either species (S1 Table).

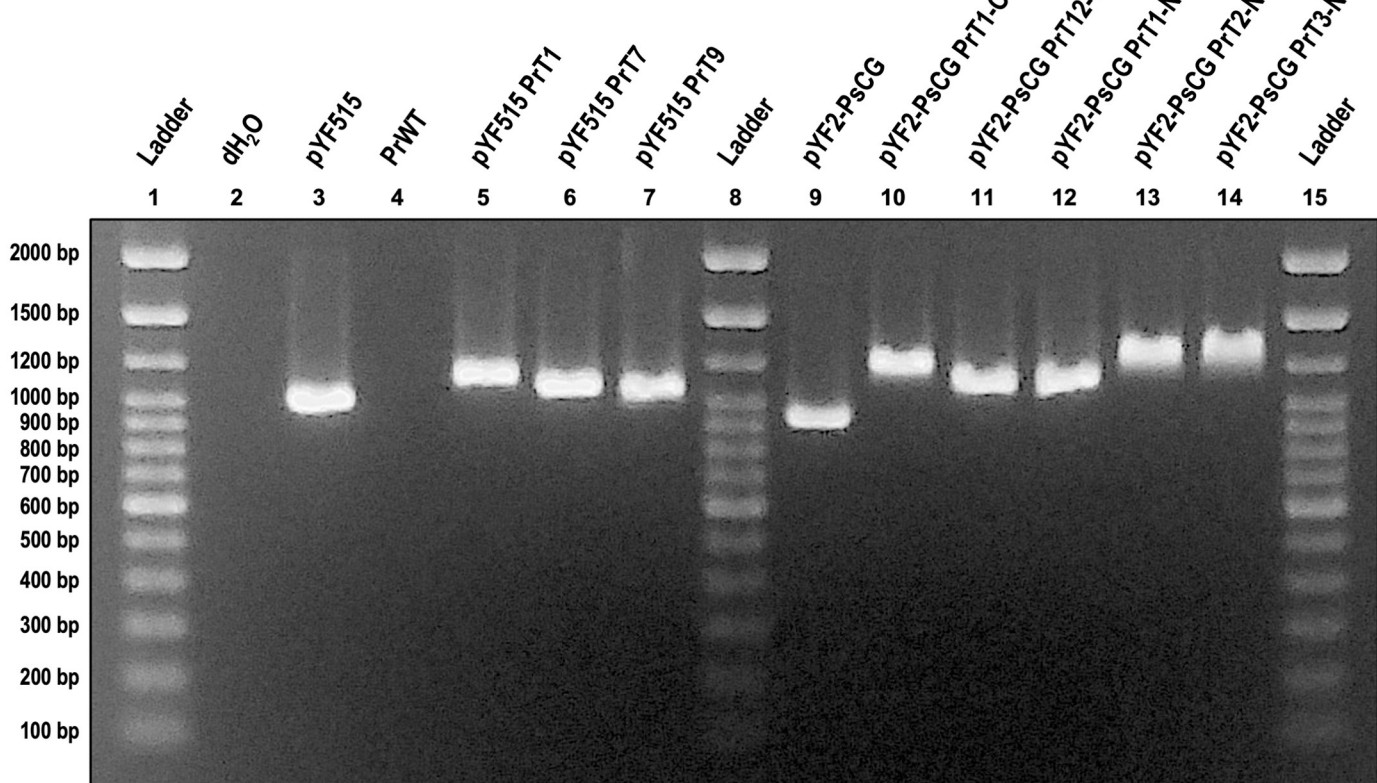

**Fig 6. Diagnostic PCR results for *Phytophthora ramorum* wildtype and plasmid (pYF2-PsCG, pYF515) transformants.** Gel electrophoresis of products from a PCR performed on DNA from *P. ramorum* NA2_17 wildtype (PrWT; well 4), pYF515 transformant (pYF515 PrT; wells 5–7), and pYF2-PsCG transformant (pYF2-PsCG PrT; wells 10–14) cultures using the pYF_Cas9_Diag_F3/R3 primer pair (expected product size is 824 bp). The negative PCR control was a reaction using molecular-grade water instead of DNA (well 2). The positive PCR controls are pYF515 plasmid DNA (well 2) and pYF2-PsCG plasmid DNA (well 9). The gel was run at 50 volts for 90 minutes on 1.25% agarose in 0.5X TBE buffer with a 100 bp DNA Ladder (Thermo Fisher Scientific, Waltham, MA, USA). All samples were loaded with Safe-Green™ stain (Applied Biological Materials Inc., Richmond, BC, Canada) at a ratio of 1 μL Safe-Green: 5 μL diluted PCR product.

## Discussion

Developing molecular techniques in Phytophthoras, and oomycetes in general, is known to be difficult and inconsistent between species [5, 25, 34, 45], and our results certainly lend evidence to this trend. Of the five forest *Phytophthora* species we tested with a PEG-CaCl₂ transformation method, only one species, *P. cactorum*, could be reliably and stably transformed with plasmid DNA. Though some initial success was achieved with *P. ramorum*, the transformants exhibited weakened and atypical growth, and subsequent transformations suddenly failed. Regarding the objectives of this study, these results have led us to conclude that first, PEG-CaCl₂ plasmid transformations are not readily adaptable between forest *Phytophthora* species, and second, that of the five species tested, *P. cactorum* is the candidate that is most amenable to CRISPR-Cas9 protocol development.

The lack of adaptability of the PEG-CaCl₂ plasmid transformation method across the five forest *Phytophthora* pathogens tested in this study likely cannot be explained by any one single factor. The use of protoplast viability controls in our experiments allows us to rule out the physical stress of transformation itself as the reason for the failed recovery of transformants in our tested species. We will therefore explore the molecular mechanisms potentially responsible for the variation in transformation success. DNA-mediated transformation involves complex

molecular pathways, and *Phytophthora* species are known for responding variably to PEG-mediated transformation protocols [25], with results sometimes even varying between isolates of the same species [36]. Our results revealed no phylogenetic pattern associated with which forest Phytophthoras could be successfully transformed. For example, transformations failed in *P. cinnamomi* even though it is closely related to the control species *P. sojae* in *Phytophthora* Clade 7 [46], but the same transformation protocol and plasmid DNA worked well for *P. cactorum* despite its phylogenetic distance in Clade 1 [46]. However, *Phytophthora* species within a clade are known to show as much variation in their biological characteristics as species between clades [47]. Comparative genomics studies have revealed that *Phytophthora* genomes are highly variable, even between closely related species [48, 49]. This biological and genomic variability could contribute to the lack of phylogeny-related similarities in response to taking up exogenous plasmid DNA.

Our success with transformations in *P. cactorum* echoes that of Chen *et al.* [27] who stably transformed a single isolate of *P. cactorum* with the pGFPN plasmid. Using the optimized PEG-mediated transformation protocol [27], Chen *et al.* obtained 25 *P. cactorum* transformants, all exhibiting green fluorescence from the *gfp* gene. Notably, Chen *et al.* transformed protoplasts with 13 pmol of DNA, which equates to 55 μg for the pGFPN plasmid and 100 μg for the pYF2-PsCG plasmid. We were able to consistently transform *P. cactorum* with 25–30 μg of plasmid DNA, suggesting that less DNA may be required for stable transformation. Taken together with previous research, our results suggest that *P. cactorum* is more amenable to DNA transformation. It could be that there is less intra-species genomic variation between isolates of *P. cactorum*, but this would need to be confirmed with comparative genomics studies. Given that *P. cactorum* is a broad host-range pathogen with impacts in horticulture, agriculture, and forestry [47, 50], it is a good candidate species for continued development of molecular techniques such as CRISPR-Cas gene editing.

The inconsistency in transformation success we observed in *P. ramorum* was somewhat unexpected given that a previous study had demonstrated stable expression of plasmid DNA in this species [36]. Riedel *et al.* used a modified version of the original PEG-liposome-mediated protocol developed in *P. infestans* [17] to transform four independent isolates of *P. ramorum* (EU1 and NA1 lineages) with 20 μg of the vector p34GFN, which contains *nptII* and *gfp* genes [24], both driven by the *B. lactucae ham34* promoter. A total of 40 transformants were obtained across all four *P. ramorum* isolates, all of which yielded positive results for p34GFN DNA in PCR and qPCR [36]. However, only five transformants from two of the four isolates exhibited *gfp* expression, indicating that the genetic background of the isolates differed enough to influence transformation results [36]. It is possible that the genotype of the *P. ramorum* NA2_17 isolate (NA2 lineage) that we used in our experiments is less amenable to transformation than the isolates used by Riedel *et al* (EU1 and NA1 lineages). Genome-wide comparisons between *P. ramorum* isolates suggest that there are hundreds of thousands years of divergence between the different lineages [51], which could certainly contribute to a variation in response to DNA-mediated transformation. Also of note is that Riedel *et al.* used Lipofectin in addition to PEG in their transformations [36]; we did try a liposome-mediated method in one of our *P. ramorum* transformations, however, we found that it had no effect. Future experiments could explore optimizing liposome reagent concentrations to determine if it improves plasmid transformations in *P. ramorum*.

Similar to *P. ramorum*, previous studies in *P. cinnamomi* were successful in obtaining stable transformants [35, 37], so our unsuccessful result in this species was unexpected. Given that Dai *et al.* [37] used an almost identical protocol to ours, based on the optimized PEG-CaCl$_2$ transformation method [25], there are two predominant explanations for our lack of transformants. The first possible explanation is that the genotype of the *P. cinnamomi* isolates we tested

may be unamenable to transformation of exogenous DNA as suggested above with *P. ramorum*. However, given that we tested two different isolates of *P. cinnamomi*, and that two previous studies successfully transformed multiple isolates of *P. cinnamomi* [35, 37], our failure to obtain transformants in this species is likely not due to the genetic background of the isolates. The second, and more likely, possible explanation is the different plasmid DNA used for transformations. Dai *et al.* [37] used a 7214 bp plasmid, pTOReGFP, which contains the *nptII* gene for G418 resistance driven by the *hsp70* promoter, while the *nptII* gene in the pYF2-PsCG plasmid we used for *P. cinnamomi* transformations is driven by the *P. sojae rpl41* promoter. It is therefore possible that the *hsp70* promoter is better suited to drive gene expression in *P. cinnamomi* than the *rpl41* promoter. The effects that differences in promoter activity and other plasmid-related factors may have on transformation success are discussed in more detail below.

An important factor to consider for transformation success is the vector DNA. Our sequencing results indicated that the pGFPN plasmid is not stable; sequences from the two independent pGFPN preparations showed recombination, inversion, and deletion of large segments of the plasmid DNA. This is particularly perplexing given that the *E. coli* strain used for plasmid replication was the laboratory standard DH5α strain with a *rec*A1-*end*A1 genotype, which should prevent unwanted recombination and cleavage in the plasmid DNA. Additionally, this instability was not observed for either pYF515 or pYF2-PsCG, which were subject to the same methods as pGFPN. The recombination events in both pGFPN preparations yielded plasmids with no *nptII* gene, thus the failed transformations with pGFPN in all tested *Phytophthora* species can confidently be attributed to the missing antibiotic selection gene. Therefore, we cannot make any meaningful conclusions regarding the efficacy of the pGFPN plasmid for transformations in forest Phytophthoras.

The only plasmid that we tested across all six *Phytophthora* species was pYF2-PsCG. One plasmid-related factor that may have influenced transformation success is variation in the activity of the *P. sojae RPL41* promoter in the five forest *Phytophthora* species. In pYF2-PsCG, the *RPL41* promoter drives expression of the *nptII* gene, and therefore any differences in promoter activity would have affected the recovery of transformants on G418-selective medium. Previous studies demonstrated that the *P. sojae RPL41* promoter is active in other *Phytophthora* species, including *P. capsici* [52], *P. litchii* (C.C. Chen ex W.H. Ko, H.S. Chang, H.J. Su, C.C. Chen & L.S. Leu) P.K. Chi, X.P. Pang & R. Liu [12], *P. infestans* [15], and *P. colocasiae* Racib [13]. However, it is possible that the *RPL41* promoter is more active in *P. cactorum* compared to the other four species we tested. Our transformations with the pGFPN plasmid could have provided more evidence for the role of promoter activity as the *nptII* gene in pGFPN is driven by the *Bremia lactucae hsp70* promoter, however, the instability of the plasmid prevented transformation. To determine the role of promoter activity in forest *Phytophthora* transformation success, future studies could test plasmids with the *nptII* gene driven by the more standard *ham34* and *hsp70* oomycete promoters, which both express well in a diversity of *Phytophthora* [17, 21–23, 25, 27, 36] and other oomycete species [32, 53, 54].

Another plasmid-related factor that could have influenced transformation success is Cas9 toxicity. Given that the pYF2-PsCG plasmid was originally constructed to test Cas9 expression [9, 42], it contains the gene for SpCas9 driven by the constitutive *ham34* promoter. While there is no single-guide RNA (sgRNA) construct in the plasmid, the Cas9 gene would still be transcribed and translated into a functional nuclease in transformed cells. Previous studies have demonstrated that stable expression of Cas9, even without a bound sgRNA or when in a catalytically deactivated form (dCas9), can be toxic in bacteria [55, 56], protozoa [57], fungi [58], and algae [59]. The exact mechanisms driving this toxicity are unclear, however, dCas9 has been shown to bind non-specifically to NGG PAM sites in the genome, even when the

nuclease is not associated with an sgRNA [60]. It has been hypothesized that this non-specific binding contributes to the toxicity of dCas9 and Cas9 when they are stably expressed in cells [61, 62]. Furthermore, previous studies trying to establish CRISPR-Cas9 gene editing in *P. infestans* have found evidence for Cas9 toxicity using the same pYF2 plasmid backbone as we used in our study [15, 63]. This apparent Cas9 toxicity has led researchers to pursue Cas12a as an alternative gene editing tool for *P. infestans* [15, 64]. In light of these previous studies, it is possible that in our study, the SpCas9 expressed from the pYF2-PsCG plasmid was toxic to *P. cinnamomi*, *P. cryptogea*, *P. syringae*, and perhaps even *P. ramorum*.

Establishing plasmid transformations in forest Phytophthoras is clearly complex, and future success will likely require tailoring protocols to each individual species, and potentially also to individual isolates. Given that in the current study, we were testing transformations in several *Phytophthora* species simultaneously, there was a limit to the breadth of protocol modifications we were able to investigate, which can be explored in future research. There are many possible directions for future studies, and our results lay the groundwork for the optimization of PEG-mediated DNA transformations in *Phytophthora* forest pathogens. The continued development of molecular techniques and species-specific protocols is necessary to improve understanding of phytopathogenicity in Phytophthoras and to ultimately develop more effective management approaches.

## Supporting information

**S1 Appendix. Supplementary materials and methods.** Details of the recipes for all culture media and transformation solutions used in this study.
(DOCX)

**S1 Table. Protocol modifications tested for *Phytophthora* transformations.** The table summarizes the conditions tested for plasmid transformations of five forest *Phytophthora* pathogens (*P. cactorum*, *P. cinnamomi*, *P. cryptogea*, *P. ramorum*, and *P. syringae*) and a control species *P. sojae*. The control transformations in *P. sojae* are listed first followed by the transformations in the five forest *Phytophthora* organized in chronological order by species name (order in which we conducted the transformation experiments).
(XLSX)

**S1 File. pYF2-PsCG plasmid sequence.** Expected sequence (based on previously published studies) of the pYF2-PsCG plasmid in FASTA format.
(TXT)

**S2 File. pYF515 plasmid sequence.** Expected sequence (based on previously published studies) of the pYF515 plasmid in FASTA format.
(TXT)

**S3 File. pGFPN plasmid sequence.** Expected sequence (based on previously published studies) of the pGFPN plasmid in FASTA format.
(TXT)

**S1 Fig. Plasmid pYF2-PsNLS-hSpCas9-GFP.** Map of the pYF2-PsNLS-hSpCas9-GFP (abbreviated pYF2-PsCG) transformation plasmid developed by Fang and Tyler (2016) for expression of hSpCas9 in *Phytophthora sojae*. The plasmid sequence was determined from Fang and Tyler (2016) and Fang *et al.* (2017) and is provided in FASTA format in S1 File. Plasmid map created using SnapGene® Viewer software v. 7.0.1 (from Dotmatics; available at snapgene.com).
(TIF)

**S2 Fig. Plasmid pYF515.** Map of pYF515, the 'all-in-one' plasmid developed by Fang et al. (2017) for expression of hSpCas9 and a single-guide RNA (sgRNA) for CRISPR/Cas9 gene editing. The plasmid sequence was supplied by Dr. Felipe Arredondo (Oregon State University) and is provided in FASTA format in S2 File. Plasmid map created using SnapGene® Viewer software v. 7.0.1 (from Dotmatics; available at snapgene.com).
(TIF)

**S3 Fig. Plasmid pGFPN.** Map of the transformation plasmid pGFPN developed by Ah-Fong and Judelson (2011). The plasmid sequence was supplied by Dr. Felipe Arredondo (Oregon State University) and is provided in FASTA format in S3 File. Plasmid map created using SnapGene® Viewer software v. 7.0.1 (from Dotmatics; available at snapgene.com).
(TIF)

**S4 Fig. Geneticin MIC growth assay example.** An example of the *P. cactorum* isolate Larch FF-42 2Pa growth assay demonstrating how the geneticin (G418) minimum inhibitory concentration (MIC) was determined for each forest *Phytophthora* species in this study. Each isolate was grown on V8 agar supplemented with a gradient of G418 concentrations (labels above the image), and two replicate plates were tested for each concentration (labels to the left of image). The first G418 concentration at which mycelial growth stopped was recorded as MIC for that species/isolate.
(TIF)

**S5 Fig. pYF515, pYF2-PsCG, and pGFPN plasmid sequence alignments.** A) Paired alignment of the expected pYF515 sequence (top) and the Plasmidsaurus pYF515 sequence (bottom). The pairwise identity of the two sequences is 99.8% with minor differences in sequence in the *nptII* gene and its *ham34* promoter, both the *ham34* terminators, and in the origin of replication (ori). B) Paired alignment of the expected pYF2-PsCG sequence (top) and the Plasmidsaurus pYF2-PsCG sequence (bottom). The pairwise identity of the two sequences is 99.8% with minor differences in sequence at the end of the Cas9 gene, in the *ham34* terminator, and in the origin of replication (ori). C) Paired alignment of the Plasmidsaurus pGFPN (first plasmid prep) sequence (top) and the expected pGFPN sequence (bottom). The pairwise identity of the two sequences is 52.9%, with sequence inversions, repeats, and deletion of the *nptII* gene in the Plasmidsaurus sequence (9,842 bp). D) Paired alignment of the expected pGFPN sequence (top) and the Plasmidsaurus pGFPN (second plasmid prep) sequence (bottom). The pairwise identity of the two sequences is 62.6% with sequence inversions, and a deletion of the *nptII* gene in the Plasmidsaurus sequence (4,927 bp). All four alignments were generated in Geneious Prime 2023.2.1 (https://www.geneious.com) using the Geneious assembler (Plasmidsaurus sequence mapped to expected sequence).
(TIF)

**S1 Raw image.**
(PDF)

## Acknowledgments

We would like to extend our sincere gratitude to all the scientists and organizations who provided lab training, plasmid DNA, *Phytophthora* cultures, and other research support.

## Author Contributions

**Conceptualization:** Erika N. Dort, Richard C. Hamelin.

**Data curation:** Erika N. Dort.

**Formal analysis:** Erika N. Dort.

**Funding acquisition:** Erika N. Dort, Richard C. Hamelin.

**Investigation:** Erika N. Dort.

**Methodology:** Erika N. Dort, Richard C. Hamelin.

**Project administration:** Erika N. Dort, Richard C. Hamelin.

**Resources:** Erika N. Dort, Richard C. Hamelin.

**Supervision:** Richard C. Hamelin.

**Validation:** Erika N. Dort.

**Visualization:** Erika N. Dort.

**Writing – original draft:** Erika N. Dort.

**Writing – review & editing:** Erika N. Dort, Richard C. Hamelin.

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
