## [Decision Letter · Decision Letter 0]

2 Jul 2024

PONE-D-24-23800Heterogeneity in establishment of polyethylene glycol-mediated plasmid transformations for five forest pathogenic Phytophthora speciesPLOS ONE

Dear Dr. Dort,

Thank you for submitting your manuscript to PLOS ONE. After careful consideration, we feel that it has merit but does not fully meet PLOS ONE’s publication criteria as it currently stands. Therefore, we invite you to submit a revised version of the manuscript that addresses the points raised during the review process.

We look forward to receiving your revised manuscript.

Kind regards,

Simon Francis Shamoun, Ph.D.

Academic Editor

PLOS ONE

Journal Requirements:

2. We noted in your submission details that a portion of your manuscript may have been presented or published elsewhere:

"Yes, but only in a PhD dissertation published on the University of British Columbia's digital repository (cIRcle). It is not a manuscript and was not submitted to a scientific journal, and therefore does not constitute dual publication."

Please clarify whether this publication was peer-reviewed and formally published. If this work was previously peer-reviewed and published, in the cover letter please provide the reason that this work does not constitute dual publication and should be included in the current manuscript.

4. Please note that Table 2 is missing, but Tables 1, 3, and 4 is included in the manuscript. 

5. We are unable to open your Supporting Information file S1_file.fa, S2_file.fa, and S3_file.fa. Please kindly revise as necessary and re-upload.

Reviewers' comments:

Reviewer's Responses to Questions

**Comments to the Author**

1. Is the manuscript technically sound, and do the data support the conclusions?

Reviewer #1: Partly

Reviewer #2: Yes

2. Has the statistical analysis been performed appropriately and rigorously? 

Reviewer #1: N/A

Reviewer #2: Yes

3. Have the authors made all data underlying the findings in their manuscript fully available?

Reviewer #1: Yes

Reviewer #2: Yes

4. Is the manuscript presented in an intelligible fashion and written in standard English?

Reviewer #1: Yes

Reviewer #2: Yes

5. Review Comments to the Author

Reviewer #1: Genetic transformation has been available for agricultural Phytophthoras for decades but not for forest Phytophthoras. The authors undertook a significant effort to assess the suitability of five common forest Phytophthora pathogens for this technology. My primary concern is the selection process for the isolates used in the transformation. The efficiency of genetic transformation in fungi (and likely oomycetes as well) can vary greatly among different isolates. It is a common practice to screen for isolates that exhibit both high virulence and high amenability to transformation. My specific comments and questions are detailed below.

L154 Protoplast isolation and transformation

Please provide information on the source of enzymes used in this study.

L170 After three days of growth

The growth rates of Phytophthora strains can vary, and the growth phase and physiological state of the mycelia can affect their transformation competence. Did the authors make any effort to optimize the growth duration for each species?

L241 Table 2 is missing

L254 pGEPN samples were 9,842 bp and 4,927 bp

Have the authors linearized those plasmids with an appropriate enzyme and estimated the sizes on the agarose gel? If so, were the estimated values consistent with the sequencing results?

L293 PCR performed on DNA from from (delete one)

L295 (expected product size is 232 bp).

Why do the sizes of PCR products on lanes 3 and 4 look different? Also, why do all products look equal or larger than 300 bp?

L362 Of the five forest Phytophthora species we tested ...

One or two isolates were tested for each species. Is there any research indicating that variations in transformation efficiency between species are greater than those within species? It's possible that testing eight isolates of P. cactorum could show a similar level of variation in transformation efficiency observed in this study.

L431 ... makes this explanation somewhat less likely.

Can you provide a p-value?

L471 ... Cas9 toxicity.

Cas12 might be more suitable for some Phytophthoras as demonstrated by Mendoza et al.,

https://doi.org/10.1094/MPMI-05-23-0072-SC

Reviewer #2: Dear Authors, thank you for submitting the manuscript to Plos One.

A few suggestions and comments are in the file attached.

Moreover, I think there are too many results in the supplementary materials, so it is better to move to the main text. Check the references and rightly put them as requested by the journal.

The English can be improved.

6. PLOS authors have the option to publish the peer review history of their article (what does this mean?). If published, this will include your full peer review and any attached files.

Reviewer #1: No

Reviewer #2: No

---

## [Author Response · Author response to Decision Letter 0]

23 Aug 2024

All responses to the reviewer and editor comments were uploaded in a separate "Response to Reviewers" rebuttal letter as instructed in the Decision Letter.

---

## [Editor Report · Decision Letter 1]

27 Aug 2024

Heterogeneity in establishment of polyethylene glycol-mediated plasmid transformations for five forest pathogenic Phytophthora species

PONE-D-24-23800R1

Dear Dr. Dort,

We’re pleased to inform you that your manuscript has been judged scientifically suitable for publication and will be formally accepted for publication once it meets all outstanding technical requirements.

Kind regards,

Simon Francis Shamoun, Ph.D.

Academic Editor

PLOS ONE
---

## [Editor Report · Acceptance letter]

30 Aug 2024

PONE-D-24-23800R1 

PLOS ONE

Dear Dr. Dort, 

I'm pleased to inform you that your manuscript has been deemed suitable for publication in PLOS ONE. Congratulations! Your manuscript is now being handed over to our production team.

Kind regards, 

on behalf of

Dr. Simon Francis Shamoun 

Academic Editor

PLOS ONE